# Criterion-related validity of Bedriddenness Rank with other established objective scales of ADLs, and Cognitive Function Score with those of cognitive impairment, both are easy-to-use official Japanese scales: A prospective observational study

Masaki Tago[1]*, Risa Hirata[1], Naoko E. Katsuki[1], Eiji Nakatani[2,3], Yoshimasa Oda[4], Shun Yamashita[1], Midori Tokushima[1], Yoshinori Tokushima[1], Hidetoshi Aihara[1], Motoshi Fujiwara[1], Shu-ichi Yamashita[1]

1 Department of General Medicine, Saga University Hospital, Saga, Japan, 2 Graduate School of Public Health, Shizuoka Graduate University of Public Health, Shizuoka, Japan, 3 Translational Research Center for Medical Innovation, Foundation for Biomedical Research and Innovation at Kobe, Hyogo, Japan, 4 Department of General Medicine, Yuai-Kai Foundation and Oda Hospital, Saga, Japan

* tagomas@cc.saga-u.ac.jp

# Abstract

## Aim

Bedriddenness Rank (BR) and Cognitive Function Score (CFS), issued by the Ministry of Health, Labour and Welfare, Japan, are easy-to-use and widely used in the medical and long-term care insurance systems in Japan. This study aims to clarify the criterion-related validity of the CFS with the Mini-Mental State Examination (MMSE) and ABC Dementia Scale (ABC-DS), and to re-evaluate the criterion-related validity of BR with the Barthel Index (BI) or Katz Index (KI) in more appropriate settings and a larger population compared with the previous study.

## Methods

A single-center prospective observational study was conducted in an acute care hospital in a suburban city in Japan. All inpatients aged 20 years or older admitted from October 1, 2018 to September 30, 2019. The relationship between BR and the BI and KI, and the relationship between CFS and the MMSE and ABC-DS were analyzed using Spearman's correlation coefficients.

## Results

We enrolled 3,003 patients. Of these, 1,664 (56%) patients exhibited normal BR. The median (interquartile range) values of the BI and KI were 100 (65–100) and 6 (2–6),

**Data Availability Statement:** The datasets generated and analyzed in the current study are available in the UMIN-ICDR repository, https://upload.umin.ac.jp/cgi-open-bin/ctr_e/ctr_view.cgi?recptno=R000051479.

**Funding:** Masaki Tago is supported by grants from Japan Society for the Promotion of Science, JSPS KAKENHI Grant Number JP18K17322 and JP21H03166. The sponsor of the study had no role in the study design, data collection, analysis, or preparation of the manuscript.

**Competing interests:** "The authors have declared that no competing interests exist"

**Abbreviations:** ADLs, Activities of Daily Living; BI, Barthel Index; KI, Katz Index; BR, Bedriddenness Rank; CFS, Cognitive Function Score; MHLW, Ministry of Health, Labour and Welfare; MMSE, Mini-Mental State Examination; ABC, Dementia Scale: ABC-DS; IQR, interquartile ranges.

respectively. Spearman's rank correlation coefficients between BR and the BI and KI were −0.891 ($p < 0.001$) and −0.877 ($p < 0.001$), respectively.

Of the patients, 1,967 (65.5%) showed normal CFS. The median (interquartile range) MMSE of 951 patients with abnormal CFS and ABC-DS of all patients were 15 (2–21) and 117 (102–117), respectively. Spearman's rank correlation coefficients between CFS and MMSE and ABC-DS were −0.546 ($p < 0.001$) and −0.862 ($p < 0.001$), respectively.

## Conclusions

BR and CFS showed significant criterion-related validity with well-established but complicated objective scales for assessing activities of daily living and cognitive functions, respectively. These two scales, which are easy to assess, are reliable and useful in busy clinical practice or large-scale screening settings.

## Introduction

Japan is classified as a super-aged society, and has the highest rate of aging in the world [1]. The age of the Japanese population continues to increase [2]. More than 70% of all hospitalized patients in Japan were recently reported to be aged 65 or older [3]. Thus, the number of older people requiring an official assessment of the need for long-term care in Japan has increased in recent years. Approximately 60,000 The number of aged people undergoing such kinds of assessment in 2019 was approximately 60,000, which is approximately 1.5-times more than the number 10 years ago, with 71% of these individuals requiring long-term care in their daily lives [4]. Admitting older patients can result in considerable work for healthcare staff, with complicated assessments concerning various aspects of nursing care, in addition to the need to acquire information directly related to medical treatments. Moreover, the presence of possible and highly probable decline of activities of daily living (ADLs) of such patients during hospitalization may further require daily care while in hospital, as well as care regarding medical treatment. As a super-aged society, human resources in Japan are typically limited by a dearth of young workers, increasing the workload of medical personnel [5]. In addition, the number of older patients undergoing home medical care is also increasing in Japan [3], aggravating the shortage of human resources in the medical field. Under these circumstances, it is particularly crucial to have expeditious and easy-to-use tools for assessing ADLs and cognitive function in older patients.

Most hospitals admitting large numbers of older patients consecutively assess ADLs of inpatients during hospitalization because ADLs tend to deteriorate with periods of prolonged hospitalization for older adults [6,7]. To this end, the Barthel Index (BI) [8] and Katz Index (KI) [9] are commonly used by many hospitals. In addition, screening for cognitive impairment on admission is also essential, because 12%–50% of inpatients have been reported to have dementia [10], and dementia can deteriorate with the decline of ADLs during hospitalization [11].

Bedriddenness Rank (BR) and Cognitive Function Score (CFS), issued by the Ministry of Health, Labour and Welfare (MHLW), Japan [12] in the 1990s, are widely used in the medical and long-term care insurance systems in Japan [12–15]. BR assesses the degree of bedriddenness, which is classified into nine grades using four classification steps [12,16]. CFS is used to evaluate cognitive impairment, which is classified into eight grades using five classification

steps [12,16]. However, few studies have examined the criterion-related validity of BR or CFS. In a previous study, we reported that BR, an official scale for assessing disability in daily life for older people in Japan, had high criterion-related validity with the BI [8] and KI [9], which are widely used as objective indices of ADLs internationally despite being relatively complicated and time-consuming to perform. In addition, BR had high inter-rater reliability [17]. Our results also revealed that CFS, an official scale for assessing patients with dementia, had high criterion-related validity with the BI [8] and KI [9], and high inter-rater reliability [17]. However, despite our previous reports, further appropriate validation studies may be useful because approximately 90% of participants were excluded in our previous study, potentially biasing the results [17]. In addition, there have been no reports regarding the criterion-related validity of CFS with well-established scales for cognitive impairment, which are currently widely used. The Mini-Mental State Examination (MMSE) [18] and the ABC Dementia Scale (ABC-DS), developed recently in Japan [19–21], are widely used as objective scales for cognitive impairment, and both scales are reported to have high inter-rater reliability [18,19]. However, the MMSE requires 11 questions to be assessed, as well as various equipment used for naming items and drawing a picture [18], and takes an average of 7.4 minutes; this makes it challenging to assess large numbers of inpatients in busy clinical settings [22]. The process involved in administering the ABC-DS is also more complicated than that for CFS, which is an official scale in Japan that is easy to use and requires the assessment of 13 questions about activities of daily living, behavioral and psychological symptoms of dementia, and cognitive function on a 9-point scale [19–21]. Although CFS is usually used to measure limitations of ADLs caused by dementia, it could potentially be used for assessing cognitive impairment, if significant correlations with the MMSE or ABC-DS were demonstrated.

The current study sought to evaluate the criterion-related validity of the CFS in relation to the MMSE and ABC-DS, in addition to re-evaluating the criterion-related validity of BR in relation to the BI or KI with more appropriate settings and a larger population compared with our previous study [17].

## Materials and methods

### Study design and patients

This study was a hospital-based prospective observational study. We recruited inpatients who were aged 20 years or older at Yuai-Kai Foundation and Oda Hospital (S1 Appendix), an acute care hospital in a suburban city in Japan, admitted from October 1, 2018, to September 30, 2019. Patients who met the following criteria were excluded: protocol deviation (CFS was normal despite MMSE was assessed), or suspected input errors (BR was normal despite BI < 10, or CFS was normal despite ABC-DS < 10, or BR or CFS were not assessed).

### Data and data sources

The variables we checked from the medical records on admission were age, sex (male or female), duration of hospitalization, BR and CFS [12,16], basic ADLs (eating, moving, personal maintenance, going to the toilet, bathing, walking, going up and down the stairs, dressing, defecation, and urination; independently or not) and ABC-DS [19–21]. Furthermore, we calculated BI [8] and KI [9] using data on basic ADLs that were routinely and systematically assessed in each patient on admission by an attending nurse. ABC-DS was evaluated and recorded by an attending nurse by interviewing a caregiver who accompanied the patient. Only the data of total score of ABC-DS was available.

BR results were classified into five grades, which were further divided into nine grades, as follows: normal, J (J1, J2), A (A1, A2), B (B1, B2), and C (C1, C2) [17]. CFS results were

classified into six grades, which were further divided into eight grades, as follows: normal, 1, 2 (2a, 2b), 3 (3a, 3b), 4, and M [12,16]. The detailed categories of BR and CFS were shown in S1A and S1B Table. BR and CFS were assessed within 24 hours after admission and recorded on a medical chart by attending nurses. Additionally, patients with abnormal CFS underwent the MMSE [18] within 72 hours after admission by a clinical psychologist or a medical clerk of the hospital. The results were also recorded using a medical chart.

### Statistical analysis

Continuous variables were shown as medians and interquartile ranges (IQR), while categorical variables were shown as actual numbers and percentages. The correlations of BR with BI or KI, and CFS with MMSE or ABC-DS were analyzed using Spearman's rank correlation coefficient. Although BR and CFS were analyzed as ordinal variables, CFS M was excluded from the ordinal variables because its definition differed from those of the other categories of CFS: "severe psychiatric symptoms, problematic behavior, or serious physical illness requiring specialized medical care." Statistical significance was set at $p < 0.05$. IBM SPSS Statistics (version 27 IBM, Armonk, New York, USA) was used for the statistical analyses.

### Ethical considerations

This study conformed to the ethical guidelines for medical and health research involving human subjects issued by the MHLW and the Ministry of Education, Culture, Sports, Science, and Technology in Japan. This study was approved by the research ethics committee of the Yuai-Kai Foundation and Oda Hospital (No. 20180629). The study was registered at the University Hospital Medical Information Network (UMIN) at www.umin.ac.jp (UMIN ID: UMIN000045078). Written informed consent was provided and obtained individually for each patient using the hospital's comprehensive agreement method, and anonymity of patients was protected.

## Results

A total of 3,148 inpatients were admitted during the study period. Among them, 109 were excluded because they were younger than 20 years old, and 36 were excluded because of protocol deviation or suspected input errors. Although the remaining 3,003 inpatients were enrolled, BR was not assessed in 32 patients, and CFS was not assessed in 41 patients. As a result, BR was analyzed in 2,971 patients and CFS was analyzed in 2,962 patients (Fig 1). The median age of the overall population was 77 (IQR: 63–86) years, and 1,451 patients (48.3%) were male. The median length of hospital stay was 9 (IQR: 4–16) days. A breakdown of the diseases leading to patients' hospitalization according to the International Statistical Classification of Diseases and Related Health Problems 10th revision [23] is shown in S2 Table. BR was normal in 1,664 patients (56%), and BI and KI were assessed in 2,954 patients, with median values of 100 (IQR: 65–100) and 6 (IQR: 2–6), respectively (Table 1). The Spearman's rank correlation coefficients of BR with BI, and KI were −0.891 ($p < 0.001$) and −0.877 ($p < 0.001$), respectively (Fig 2A and 2B).

CFS was normal in 1,967 patients (65.5%), and the MMSE was conducted in 951 patients among the other 1,036 patients with abnormal CFS, and ABC-DS was performed in all patients regardless of the CFS results. The median values of MMSE and ABC-DS were 15 (IQR: 2–21) and 117 (IQR: 102–117), respectively (Table 1). Spearman's rank correlation coefficients of CFS with MMSE, and ABC-DS were −0.546 ($p < 0.001$) and −0.862 ($p < 0.001$), respectively (Fig 3A and 3B). Additionally, we conducted a sensitivity analysis focusing on the group of patients, for whom MMSE was evaluated because their CFS was abnormal, to improve

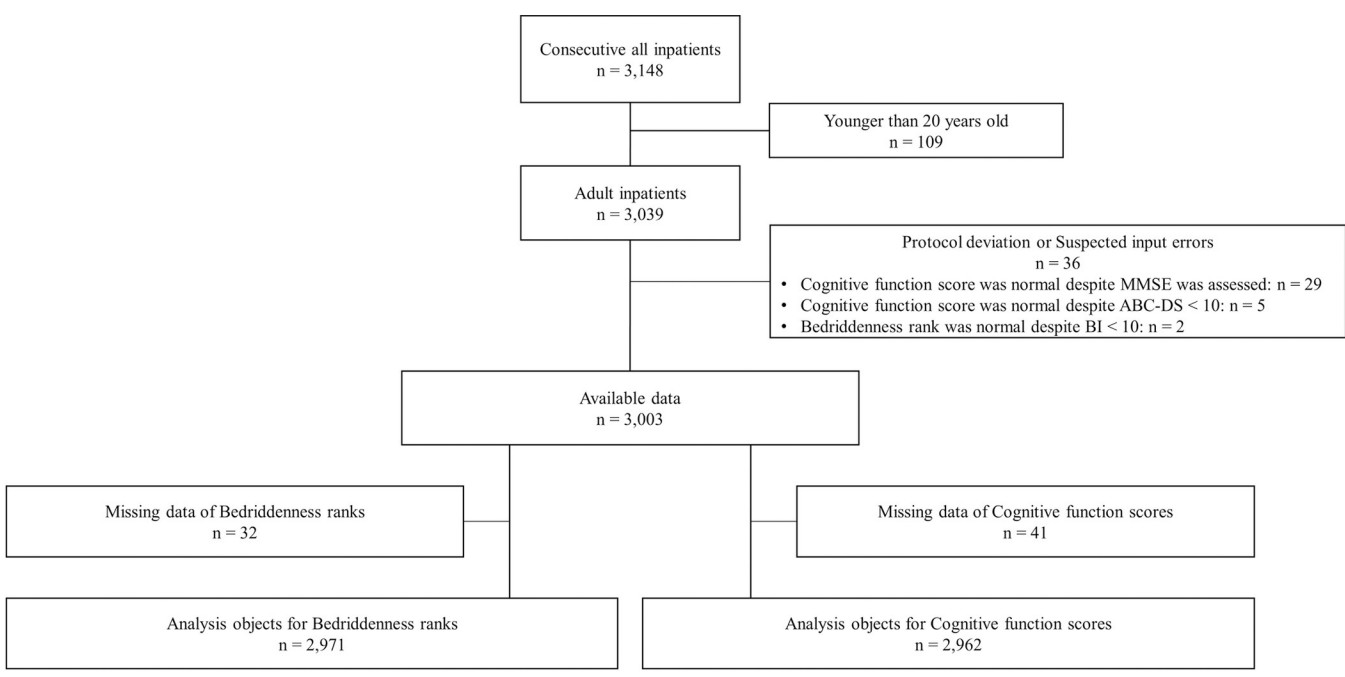

**Fig 1. Data flow diagram.**

comparability of the validation analyses between CFS and MMSE, and CFS and ABC-DS. As a result, Spearman's rank correlation coefficient of CFS with those patients was −0.698 (p < 0.001).

## Discussion

The current results revealed high criterion-related validity of BR with the BI [8] and KI [9], which are well-established objective ADL scales. In addition, we found high criterion-related validity of CFS with the MMSE [18] and ABC-DS [19–21], which are well-established objective scales for assessing cognitive impairment. Importantly, these findings indicate that it is appropriate for healthcare workers in extremely busy clinical settings to implement easy-to-use official Japanese scales (BR and CFS) to assess ADLs and cognitive impairment among older people, instead of well-established but complicated and time-consuming scales. A substantial amount of data regarding BR and CFS has already been accumulated through wide and extensive usage in Japanese medical and long-term care insurance systems as national official scales [12–15]. The current findings suggest that these data represent a valuable resource for the future study of ADLs and cognitive impairment in older people. Importantly, the current results could substantially expedite future research in geriatric and gerontological medicine, especially in Japan, which is a super-aged society with the highest rate of aging in the world [1].

A previous community-based study of 1,200 Japanese residents demonstrated the presence of a significant correlation between BR and BI [24]. The current study extended these findings, demonstrating that BR had a strong correlation with BI and KI in inpatients of an acute care hospital. In this study, patients with BR results of normal to J2 had almost the same BI (median: 95 to 100) (Fig 2A) and almost the same KI (median: 5 to 6) (Fig 2B). This finding indicates that BR could provide more detailed assessment and classification of ADLs than BI or KI for patients who are nearly independent. On the other hand, all of the categories of

**Table 1. Characteristics of enrolled patients.**

|  | Median (interquartile range) or n (%) |
|---|---|
| Age (years) | 77 (63–86) |
| Gender, Male | 1,451 (48.3) |
| Length of hospital stay (days) | 9 (4–16) |
| **Bedriddenness Rank** |  |
| Normal | 1,664 (56.0) |
| J1 | 112 (3.8) |
| J2 | 123 (4.1) |
| A1 | 160 (5.4) |
| A2 | 155 (5.2) |
| B1 | 208 (7.0) |
| B2 | 173 (5.8) |
| C1 | 127 (4.3) |
| C2 | 249 (8.4) |
| Missing | 32 (1.1) |
| **Cognitive Function Score** |  |
| Normal | 1,967 (65.5) |
| I | 334 (11.1) |
| IIa | 104 (3.5) |
| IIb | 164 (5.5) |
| IIIa | 305 (10.2) |
| IIIb | 41 (1.4) |
| IV | 42 (1.4) |
| M | 5 (0.2) |
| Missing | 41 (1.4) |
| BI, n = 2,954 | 100 (65–100) |
| KI, n = 2,954 | 6 (2–6) |
| MMSE, n = 951* | 15 (2–21) |
| ABC-DS, n = 3,003 | 117 (102–117) |

**Abbreviations:** BI: Barthel Index; KI: Katz Index; MMSE: Mini-Mental State Examination; ABC Dementia Scale: ABC-DS.

* Patients with abnormal Cognitive Function Score underwent the MMSE.

patients belonging to B2 to C2 assessed by BR had a median KI of 0. This finding indicates that BR could provide more detailed assessment and classification of ADLs than KI for patients who are nearly bedridden. Administering the BI involves the evaluation of 10 basic ADLs, such as eating, dressing, bathing, transferring, and toileting, by classifying each activity into two to four grades [8], and takes an average of 5 ± 2.58 minutes [25]. Administering the KI requires the evaluation and classification of six basic ADLs into two categories: independent or requiring assistance [9]. Therefore, both scales are complicated and time-consuming to evaluate, despite being well-established and reliable. In contrast, BR only requires asking the patient or their family members up to four steps of questions concerning their mobility [16,17], which makes the assessment very quick and easy to perform. In addition, BR exhibited high inter-rater reliability in our previous study [17], indicating that it is a useful scale for assessing ADLs in busy clinical settings.

The current study revealed strong criterion-related validity between CFS and each of the well-established scales for cognitive impairment, MMSE and ABC-DS. CFS was not originally

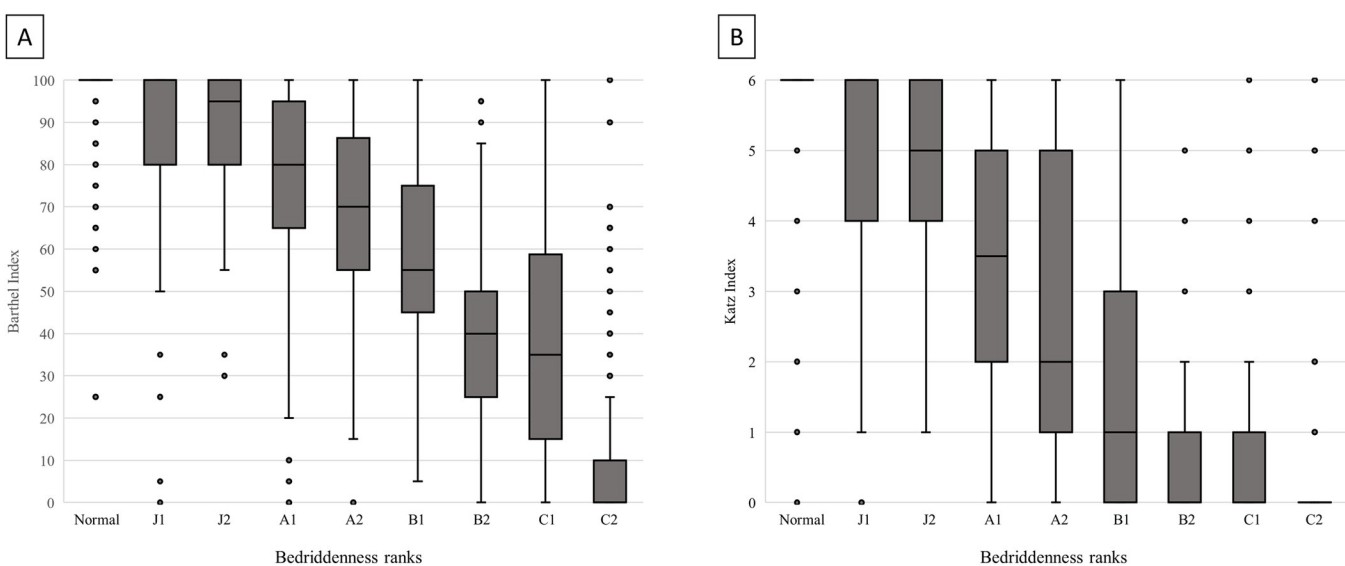

**Fig 2.** Box and whisker diagrams showing the relationship between BR and (A) BI, and (B) KI. Spearman's rank correlation coefficient between Bedriddenness Rank and Barthel Index was −0.891, p < 0.001 (A), and that between Bedriddenness Rank and Katz Index was −0.877, p < 0.001 (B).

developed to diagnose dementia or evaluate cognitive impairment, but rather to evaluate disabilities in ADLs of older patients caused by dementia, irrespective of a causative disease [26]. Although our previous study showed high inter-rater reliability and excellent criterion-related validity of CFS with BI and KI [17], to the best of our knowledge, no studies have examined the criterion-related validity of CFS with other well-established objective scales for dementia.

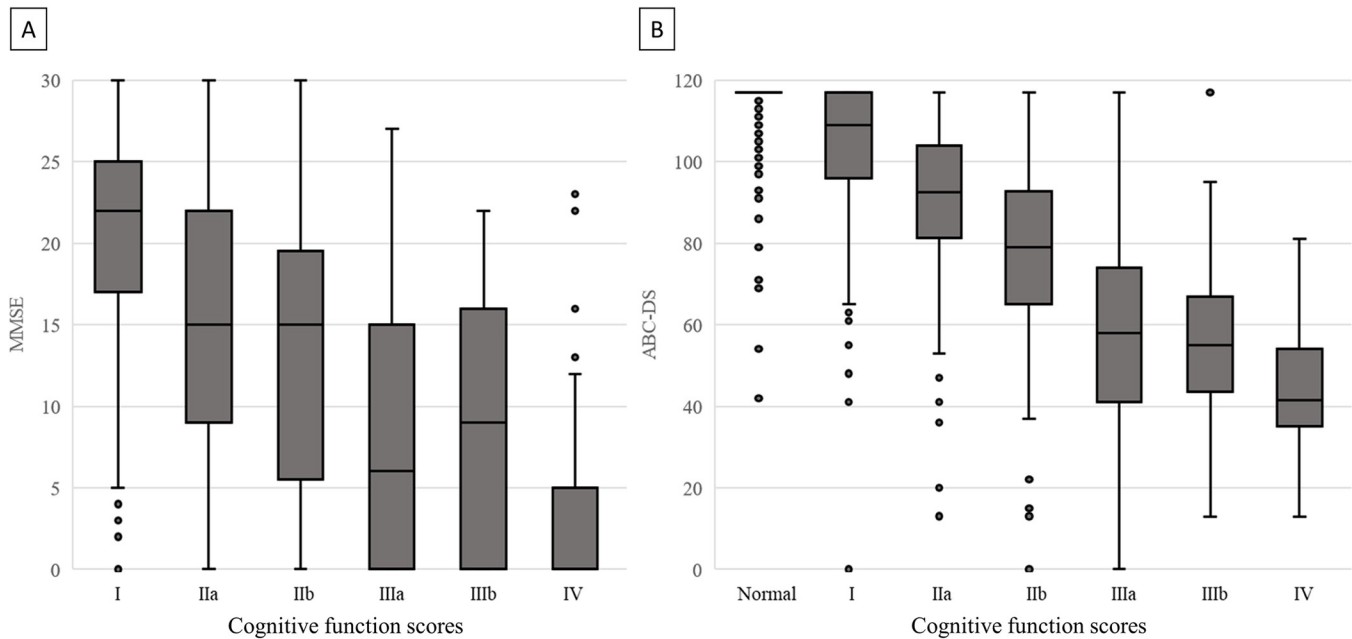

**Fig 3.** Box and whisker diagrams showing the relationship between CFS and (A) MMSE, and (B) ABC-DS. The Spearman's rank correlation coefficient between Cognitive Function Score and Mini-Mental State Examination was −0.546, p < 0.001 (A), and that between Cognitive Function Score and ABC Dementia Scale was −0.862, p < 0.001 (B).

Therefore, the results of the current study have important implications. Comparing the correlation of CFS with ABC-DS and with MMSE, the current findings indicated that the correlation with the ABC-DS (r = −0.862) was stronger than that with the MMSE (r = −0.546). Furthermore, Spearman's rank correlation coefficients of CFS with ABC-DS in the group of patients for whom MMSE was evaluated because their CFS was abnormal, revealed a strong negative correlation, with a correlation coefficient of r = −0.698 (p < 0.001). This discrepancy was likely to have been caused by the fact that ABC-DS included assessments of ADLs [21], whereas the MMSE mainly assessed cognitive functions [18]. As mentioned above, the CFS was originally developed as a scale for assessing disabilities in ADLs caused by dementia in older patients. The MMSE has been reported to have high inter-rater reliability [18] and has been widely used in clinical practice internationally [22]. Meanwhile, the ABC-DS has also been reported to have high inter-rater reliability, assessing 13 items regarding ADLs, behavioral and psychological symptoms of dementia, and cognitive functions by rating them on a scale of 1 to 9 [19]. In addition, for both the MMSE and ABC-DS, lower scores indicate more severe cognitive impairment [21,27]. The current study revealed that all of the first quartile scores of MMSE of patients with CFS from IIIa to IV were 0 (Fig 3A), suggesting that the CFS reveal more detailed assessment and classification of patients with extremely severe dementia with MMSE scores close to 0. However, the current findings also showed that the median ABC-DS score of patients with CFS belonging to IV was 40 (Fig 3B), suggesting that CFS could not provide appropriate assessment and classification of patients with severe dementia with ABC-DS below 40, which could have been caused by the ABC-DS requiring more detailed assessments in multiple dimensions apart from the evaluation of cognitive functions. However, considering that severe dementia is expressed by a score of 13 to 70 by ABC-DS [21], the CFS appears to be sufficiently suitable for evaluating cognitive impairment in actual clinical practice, even if the scale cannot classify severe dementia in detail with a score lower than 40 as assessed by the ABC-DS. Moreover, the MMSE and ABD-DS take an average of 7.4 minutes [22] and 9.9 minutes to perform [20], respectively. In contrast, the CFS requires a maximum of only five steps to make a judgment [16,17], making it a useful, quick, and easy-to-use assessment tool.

## Limitations

Although we analyzed data from a much larger population compared with our previous study, this was still a single-center study. Thus, the current study may have been affected by population bias caused by the limited number of hospitals, clinical departments, kinds of diseases, and unique characteristics of the region to which the hospital belonged. Additionally, we could not evaluate the MMSE in patients with normal CFS because of a lack of human resources, which made it impossible to clarify the relationship between normal CFS and MMSE in this study. In addition, the time required for the evaluation of BR or CFS, the simplicity of the procedures, and the validity of the assessment method of ABC-DS, were not evaluated in this study.

## Conclusion

BR showed significant criterion-related validity with well-established and objective but complicated and time-consuming scales of ADLs. In addition, CFS showed significant criterion-related validity with well-established but complicated and time-consuming cognitive impairment scales. These official Japanese scales, which are reliable and easy to perform, are useful in busy clinical practice or large-scale screening settings.

## Supporting information

**S1 Appendix. Characteristics of Yuai-Kai Foundation and Oda Hospital.**
(PDF)

**S1 Table.** Table A. The categories of Bedriddenness Ranks. Table B. The categories of Cognitive Function Scores.
(PDF)

**S2 Table. The breakdown of the diseases leading to patients' hospitalization according to the International Statistical Classification of Diseases and Related Health Problems 10th revision (ICD-10).**
(PDF)

## Acknowledgments

We thank Fujiko Eguchi, Chieko Nagaike, Kenta Yamaguchi, Yasuhiro Chibu, Osamu Kojiro, and Toshinobu Eguchi from the Yuai-Kai Foundation and Oda Hospital for assistance with data acquisition. We thank Translational Research Center for Medical Innovation and Dr. Masanori Fukushima from the Learning Health Society Institute, for research support and assistance with analysis. We thank Benjamin Knight, MSc., from Edanz (https://jp.edanz.com/ac) for editing a draft of this manuscript.

## Author Contributions

**Conceptualization:** Masaki Tago, Naoko E. Katsuki.

**Data curation:** Risa Hirata, Naoko E. Katsuki, Shun Yamashita.

**Formal analysis:** Naoko E. Katsuki, Eiji Nakatani.

**Funding acquisition:** Masaki Tago.

**Investigation:** Risa Hirata, Yoshimasa Oda, Midori Tokushima, Yoshinori Tokushima, Hidetoshi Aihara, Motoshi Fujiwara, Shu-ichi Yamashita.

**Methodology:** Masaki Tago, Naoko E. Katsuki, Eiji Nakatani, Yoshimasa Oda, Shun Yamashita.

**Project administration:** Masaki Tago.

**Resources:** Risa Hirata, Yoshimasa Oda.

**Supervision:** Masaki Tago, Eiji Nakatani, Shu-ichi Yamashita.

**Validation:** Masaki Tago, Risa Hirata, Naoko E. Katsuki, Eiji Nakatani, Yoshimasa Oda, Shun Yamashita, Midori Tokushima, Yoshinori Tokushima, Hidetoshi Aihara, Motoshi Fujiwara, Shu-ichi Yamashita.

**Writing – original draft:** Masaki Tago, Risa Hirata.

**Writing – review & editing:** Naoko E. Katsuki, Eiji Nakatani, Yoshimasa Oda, Shun Yamashita, Midori Tokushima, Yoshinori Tokushima, Hidetoshi Aihara, Motoshi Fujiwara, Shu-ichi Yamashita.

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
