## [Decision Letter · Decision Letter 0]

24 Aug 2022

PONE-D-22-10733Criterion-related validity of Bedriddenness Rank with other established objective scales of ADLs, and Cognitive Function Score with those of cognitive impairment, both are easy-to-use official Japanese scales: a prospective observational studyPLOS ONE

Dear Dr. Tago,

Thank you for submitting your manuscript to PLOS ONE. After careful consideration, we feel that it has merit but does not fully meet PLOS ONE’s publication criteria as it currently stands. Therefore, we invite you to submit a revised version of the manuscript that addresses the points raised during the review process.

The manuscript has been evaluated by two reviewers, and their comments are available below.

The reviewers have raised a number of major concerns, including the need for further details regarding key concepts, diagnoses, and methodology.  The reviewers also note concerns about the analyses presented and request that further analyses be completed.

Could you please carefully revise the manuscript to address all comments raised?

We look forward to receiving your revised manuscript.

Kind regards,

Steve Zimmerman, PhD

Associate Editor, PLOS ONE

Journal Requirements:

Masaki Tago is supported by grants from Japan Society for the Promotion of Science, JSPS KAKENHI Grant Number JP18K17322 and JP21H03166. The sponsor of the study had no role in the study design, data collection, analysis, or preparation of the manuscript.

Please respond by return email with your amended Competing Interests Statement and we will change the online submission form on your behalf.

Reviewers' comments:

Reviewer's Responses to Questions

**Comments to the Author**

1. Is the manuscript technically sound, and do the data support the conclusions?

Reviewer #1: Yes

Reviewer #2: No

2. Has the statistical analysis been performed appropriately and rigorously? 

Reviewer #1: Yes

Reviewer #2: No

3. Have the authors made all data underlying the findings in their manuscript fully available?

Reviewer #1: Yes

Reviewer #2: Yes

4. Is the manuscript presented in an intelligible fashion and written in standard English?

Reviewer #1: Yes

Reviewer #2: No

5. Review Comments to the Author

Reviewer #1: The proposed work seeks to provide further validation of the Bedriddenness Rank (BR) as a measure of ADL limitations, previously validated in a smaller study, as well as to assess validity of the Cognitive Function Score (CFS) as a measure of cognitive impairment. While I commend the authors for a well-written manuscript, I would encourage them to consider the following suggestions to further improve their work:

1. Provide a better description of BR and CFS, particularly what each category (e.g., J1, J2 for BR and I, IIa etc. for CFS) indicates. Although this information is available from prior publications of the group, access to it within the manuscript would save the reader time they would otherwise have to spend searching for it.

2. Lines 169-170 suggest that the CFS M category was defined in this study as ‘severe psychiatric symptoms, problematic behavior, or serious physical illness requiring medical care’, while in the previous validation study (Tago M et al., 2021; doi: https://doi.org/10.1186/s12877-0021-02108-x) the same category was defined as ‘Does the patient have delirium or self-inflicted harm?’. A better description of CFS (see point #1) would help clarify this confusion.

3. How long does CFS take to administer? The authors imply that some of the other available assessment tools (e.g., MMSE) are lengthy and require certain equipment to administer. However, at only 7.4 minutes average time and requiring paper and a few objects (for naming) that can be anything that is available to the rater, MMSE is generally considered a quick test that does not require special training or equipment and is widely used in the clinical setting due in part to its ease of administration. I would downplay this as a main rationale for assessing validity of CFS as a cognitive assessment.

4. For the CFS-ABC-DS analysis, consider doing a sensitivity analysis focused on the 951 patients who underwent MMSE testing. This group was more cognitively impaired (median MMSE score = 15, which indicates moderate dementia) compared to the sample of 3,003 patients with data on ABC-DS (median ABC-DS score = 117, which is equivalent to a global CDR of 0/0.5 indicating normal cognition/very mild cognitive impairment (Mori T et al., 2018; doi: 10.1159/000486956). This is likely due to the inclusion of patients with normal CFS scores in the ‘ABC-DS’ sample while they were excluded from the ‘MMSE’ sample. Assessing correlation with the two ‘gold-standard’ tests in the same sample would improve comparability of the two CFS validation analyses. Since MMSE was not done in patients with normal CFS scores, how about removing them for the ABC-DS as well? Alternatively, the CFS-MMSE correlation analysis could be repeated including the 29 patients who were excluded due to protocol deviation (CFS normal despite MMSE assessed), although I would consider doing this as a second choice given the small number of patients with normal CFS scores that would be added to the ‘MMSE’ sample.

5. I encourage the authors to consider validating CFS against the cognitive domain of ABC-DS in addition to the global score. That would provide a more direct assessment of CFS’s validity as a cognitive impairment measure and would be more consistent with the validation against MMSE.

6. Finally, could the authors comment on how ABC-DS was administered? Was the rating done by nurses/medical clerks? If yes, how valid would it be since the original scale was developed using input from caregivers who assist the patients 3 or more days a week (see the Mori article)?

Reviewer #2: First of all, I would like to apologize for the delay in reviewing the manuscript. Generally, I try to do it as fast as possible but I was unable to because of an unexpectedly busy schedule in the last month.

The current study investigates the criterion-related validity of Bedriddenness Rank (BR) and Cognitive Function Score (CFS) in subjects aged 20 years and older who were hospitalized in a suburban city in Japan. The major concern is that the subjects in the current study were probably hospitalized for a wide range of disorders that may differently affect ADLs and cognitive function. Meanwhile, it is recommended to investigate the validity and reliability of instruments/questionnaires in specific populations. In addition, no information about the reason for the hospitalization of the participants is provided in the manuscript.

Further, although the main focus of the introduction is the aging population in Japan as a super-aged society and the need to assess ADLs and cognitive function in hospitalized elderly subjects using brief and easy-to-use instruments, the authors recruited hospitalized subjects aged 20 years or older.

Another important issue is the low necessity to do such a study. As mentioned in the introduction, the criterion-related validity of the BR and CFS is re-evaluated in the current study. Further, as mentioned by the authors, CFS is usually used to measure the limitations of ADLs caused by dementia and it does not assess cognitive function. So, merely the significant correlation of CFS with MMSE or ABC-DS does not justify the use of CFS for assessing cognitive impairment instead of MMSE or ABC-DS.

6. PLOS authors have the option to publish the peer review history of their article (what does this mean?). If published, this will include your full peer review and any attached files.

Reviewer #1: No

Reviewer #2: No

---

## [Author Response · Author response to Decision Letter 0]

27 Sep 2022

Response: 

We revised the manuscript in accord with the template provided by PLOS ONE.

Response: 

We confirmed that all of the required details were included in our manuscript.

Masaki Tago is supported by grants from Japan Society for the Promotion of Science, JSPS KAKENHI Grant Number JP18K17322 and JP21H03166. The sponsor of the study had no role in the study design, data collection, analysis, or preparation of the manuscript.

Please respond by return email with your amended Competing Interests Statement and we will change the online submission form on your behalf.

Response: 

We changed the corresponding sentences in the Competing Interests Statement as instructed (as described below), which we made sure to include in our return e-mail.

Masaki Tago is supported by grants from Japan Society for the Promotion of Science, JSPS KAKENHI Grant Number JP18K17322 and JP21H03166. The sponsor of the study had no role in the study design, data collection, analysis, or preparation of the manuscript. This does not alter our adherence to PLOS ONE policies on sharing data and materials.

Response: 

We deleted unnecessary ethics statements in sections other than the Methods.

Reviewers' comments:

Reviewer's Responses to Questions

Comments to the Author

1. Is the manuscript technically sound, and do the data support the conclusions?

Reviewer #1: Yes

Reviewer #2: No

2. Has the statistical analysis been performed appropriately and rigorously?

Reviewer #1: Yes

Reviewer #2: No

3. Have the authors made all data underlying the findings in their manuscript fully available?

Reviewer #1: Yes

Reviewer #2: Yes

4. Is the manuscript presented in an intelligible fashion and written in standard English?

Reviewer #1: Yes

Reviewer #2: No

5. Review Comments to the Author

Response to the comments of Reviewer 1

Reviewer #1: The proposed work seeks to provide further validation of the Bedriddenness Rank (BR) as a measure of ADL limitations, previously validated in a smaller study, as well as to assess validity of the Cognitive Function Score (CFS) as a measure of cognitive impairment. While I commend the authors for a well-written manuscript, I would encourage them to consider the following suggestions to further improve their work:

Response: 

We appreciate Reviewer 1’s assessment of our manuscript as “well-written.” Our responses to Reviewer 1’s comments are provided below.

1. Provide a better description of BR and CFS, particularly what each category (e.g., J1, J2 for BR and I, IIa etc. for CFS) indicates. Although this information is available from prior publications of the group, access to it within the manuscript would save the reader time they would otherwise have to spend searching for it.

Response: 

In accord with Reviewer 1’s recommendation, we have described the details of the categories of Bedriddenness Ranks (BR) and Cognitive Function Scores (CFS) in S1, Table, to make the information more accessible to readers. Additionally, we added a sentence to lines 159 to 160 on page 12, as follows.

Line 159–160,

The detailed categories of BR and CFS were shown in S1, Tables A and B.

2. Lines 169-170 suggest that the CFS M category was defined in this study as ‘severe psychiatric symptoms, problematic behavior, or serious physical illness requiring medical care’, while in the previous validation study (Tago M et al., 2021; doi: https://doi.org/10.1186/s12877-0021-02108-x) the same category was defined as ‘Does the patient have delirium or self-inflicted harm?’. A better description of CFS (see point #1) would help clarify this confusion.

Response: 

We apologize for the confusion regarding the definition of category M of CFS in the previous version of our manuscript. The question, “Does the patient have delirium or self-inflicted harm?”, which was included in our previous research, was not the precise definition of category M, but a simplified criterion to facilitate the work of making a judgment in a busy medical setting. We would like to emphasize that the precise definition of category M is “severe psychiatric symptoms, problematic behavior, or serious physical illness requiring specialized medical care.” as reported in our original manuscript. The definition of category M is clearly described in S1, Table, which we added to address a previous comment by Reviewer 1.

3. How long does CFS take to administer? The authors imply that some of the other available assessment tools (e.g., MMSE) are lengthy and require certain equipment to administer. However, at only 7.4 minutes average time and requiring paper and a few objects (for naming) that can be anything that is available to the rater, MMSE is generally considered a quick test that does not require special training or equipment and is widely used in the clinical setting due in part to its ease of administration. I would downplay this as a main rationale for assessing validity of CFS as a cognitive assessment.

Response: 

We appreciate Reviewer 1’s comment. CFS is an extremely easy-to-use tool, which requires less than 2 minutes to perform with a maximum of five steps, as we already mentioned in the background section. However, we did not measure the precise length of time necessary for the evaluation of CFS. Therefore, we added a sentence to clarify that we did not evaluate the time necessary for the evaluation of CFS in this study in the Limitations section in lines 306 to 308 on page 22, as described below.

Line 306-308,

In addition, the time required for the evaluation of BR or CFS, the simplicity of the procedures, and the validity of the assessment method of ABC-DS, were not evaluated in this study.

However, we do not agree that Mini-Mental State Examination (MMSE) is a sufficiently simple and quick test. In busy clinical settings in Japan, a required duration of 7.4 minutes is too long to be useful for diagnosing dementia for a single outpatient in a clinic overloaded with large numbers of older patients. In addition, in current and actual clinical settings in Japan, medical staff with experience conducting the MMSE are often unavailable, mainly for economic reasons. Moreover, the MMSE requires the preparation of materials such as paper, pens, and evaluation sheets. Thus, we believe that the MMSE involves a substantially more complicated procedure than CFS.

4. For the CFS-ABC-DS analysis, consider doing a sensitivity analysis focused on the 951 patients who underwent MMSE testing. This group was more cognitively impaired (median MMSE score = 15, which indicates moderate dementia) compared to the sample of 3,003 patients with data on ABC-DS (median ABC-DS score = 117, which is equivalent to a global CDR of 0/0.5 indicating normal cognition/very mild cognitive impairment (Mori T et al., 2018; doi: 10.1159/000486956). This is likely due to the inclusion of patients with normal CFS scores in the ‘ABC-DS’ sample while they were excluded from the ‘MMSE’ sample. Assessing correlation with the two ‘gold-standard’ tests in the same sample would improve comparability of the two CFS validation analyses. Since MMSE was not done in patients with normal CFS scores, how about removing them for the ABC-DS as well? Alternatively, the CFS-MMSE correlation analysis could be repeated including the 29 patients who were excluded due to protocol deviation (CFS normal despite MMSE assessed), although I would consider doing this as a second choice given the small number of patients with normal CFS scores that would be added to the ‘MMSE’ sample.

Response: 

Following the recommendation of Reviewer 1, we calculated Spearman’s correlation coefficient between CFS and the ABC Dementia Scale (ABC-DS), among the patients for whom MMSE was assessed because their CFS was abnormal. The correlation coefficient showed a strong negative correlation between the two scales with a correlation coefficient of r = −0.698 (p < 0.001), which we described in lines 216 to 219 on page 17, as described below.

Line 216-219,

Additionally, Spearman’s rank correlation coefficients of CFS with ABC-DS in the group of patients for whom MMSE was evaluated because their CFS was abnormal, revealed a strong negative correlation, with a correlation coefficient of r = −0.698 (p < 0.001).

5. I encourage the authors to consider validating CFS against the cognitive domain of ABC-DS in addition to the global score. That would provide a more direct assessment of CFS’s validity as a cognitive impairment measure and would be more consistent with the validation against MMSE.

Response: 

We believe that it is acceptable to compare the total score of ABC-DS with MMSE, because its parallel validity with MMSE has been confirmed in previous studies. In addition, it was impossible to evaluate the point suggested by Reviewer 1 because of the unfortunate lack of results for each domain (A: ADL, B: BPSD, and C: Cognitive function). We appreciate the reviewer’s understanding.

6. Finally, could the authors comment on how ABC-DS was administered? Was the rating done by nurses/medical clerks? If yes, how valid would it be since the original scale was developed using input from caregivers who assist the patients 3 or more days a week (see the Mori article)?

Response: 

In the current study, attending nurses evaluated and recorded ABC-DS by interviewing caregivers who accompanied the patient. We added a sentence to clarify this point in lines 154 to 155 on page 11, as described below.

Lines 154-155,

ABC-DS was evaluated and recorded by an attending nurse by interviewing a caregiver who accompanied the patient.

In addition, as Reviewer 1 mentioned, we did not confirm whether the caregivers assisted the patients for 3 or more days a week. Therefore, we added a sentence to describe this point in the Limitations section in lines 306 to 308 on page 22, as described below.

Line 306-308,

In addition, the time required for the evaluation of BR or CFS, the simplicity of the procedures, and the validity of the assessment method of ABC-DS, were not evaluated in this study.

Response to the comments of Reviewer 2

Reviewer #2: First of all, I would like to apologize for the delay in reviewing the manuscript. Generally, I try to do it as fast as possible but I was unable to because of an unexpectedly busy schedule in the last month.

Response: 

We thank Reviewer 2 for reviewing our manuscript. Our responses to Reviewer 2’s comments are provided below.

1. The current study investigates the criterion-related validity of Bedriddenness Rank (BR) and Cognitive Function Score (CFS) in subjects aged 20 years and older who were hospitalized in a suburban city in Japan. The major concern is that the subjects in the current study were probably hospitalized for a wide range of disorders that may differently affect ADLs and cognitive function. Meanwhile, it is recommended to investigate the validity and reliability of instruments/questionnaires in specific populations. In addition, no information about the reason for the hospitalization of the participants is provided in the manuscript.

Response: 

We appreciate Reviewer 2’s helpful comment. We added ICD-10 classification of the diseases that led to patients’ hospitalization as S2, Table, except for data for age, gender, and length of hospital stay as inpatient information, which we had already described in our original manuscript. We added a sentence to describe this point in lines 193 to 195 on page 14, as described below.

Lines 193-195,

A breakdown of the diseases leading to patients’ hospitalization according to the International Statistical Classification of Diseases and Related Health Problems 10th revision [23] is shown in S2, Table.

We would like to emphasize that the inter-rater reliability of the BR and CFS were already confirmed in a previous study, although we did not evaluate the validity and reliability of the various instruments in the current study, as mentioned by Reviewer 2. In addition, we do not believe that evaluation of the validity and reliability of the Barthel index (BI), Katz index (KI), or MMSE are necessary, because they have been widely used in numerous studies as well-established objective instruments. However, because we failed to validate the ABC-DS assessment methods, as already mentioned in our response to Reviewer 1’s comment, we added a sentence to describe this point in the Limitations section in lines 306 to 308 on page 22, as described below.

Lines 306-308,

In addition, the time required for the evaluation of BR or CFS, the simplicity of the procedures, and the validity of the assessment method of ABC-DS, were not evaluated in this study.

2. Further, although the main focus of the introduction is the aging population in Japan as a super-aged society and the need to assess ADLs and cognitive function in hospitalized elderly subjects using brief and easy-to-use instruments, the authors recruited hospitalized subjects aged 20 years or older.

Response: 

First, we already demonstrated the correlations between BR and BI, BR and KI, CFS and BI, and CFS and KI for older patients, respectively, in our previous study. The current study involved adult patients that were 20 years of age or older. However, the median age of the study population was 77 years, because this study was conducted in the hospital in which most of the admitted patients were older people, causing no logical inconsistency in our manuscript.

Second, following Reviewer 1’s recommendation, we also calculated Spearman’s correlation coefficients between CFS and the ABC Dementia Scale (ABC-DS), among patients for whom MMSE was assessed because their CFS was abnormal. As a result, the correlation coefficient showed a strong negative correlation between those two scales with a correlation coefficient of r = −0.698 (p < 0.001), which meant that we succeeded in the validation regarding CFS in aged patients with dementia. We added a sentence to demonstrate this point at lines 216 to 219 on page 17, as described below. We hope that this resolves Reviewer 2’s concern.

Line 216-219,

Additionally, Spearman’s rank correlation coefficients of CFS with ABC-DS in the group of patients for whom MMSE was evaluated because their CFS was abnormal, revealed a strong negative correlation, with a correlation coefficient of r = −0.698 (p < 0.001).

3. Another important issue is the low necessity to do such a study. As mentioned in the introduction, the criterion-related validity of the BR and CFS is re-evaluated in the current study. Further, as mentioned by the authors, CFS is usually used to measure the limitations of ADLs caused by dementia and it does not assess cognitive function. So, merely the significant correlation of CFS with MMSE or ABC-DS does not justify the use of CFS for assessing cognitive impairment instead of MMSE or ABC-DS.

Response: 

We respectfully disagree with Reviewer 2’s comment on this point. The fact that the present study revealed a significant correlation between CFS and MMSE or ABC-DS indicates that CFS had the power to predict MMSE and ABC-DS scores. Furthermore, we propose that CFS has the potential to be used as an objective dementia scale considering the results of previous research and the present study, because the inter-rater reliability of CFS has already been established in a previous study [17]. We described this point in lines 129 to 132 on page 10, as described below.

Lines 129-132,

Although CFS is usually used to measure limitations of ADLs caused by dementia, it could potentially be used for assessing cognitive impairment, if significant correlations with the MMSE or ABC-DS were demonstrated.

In addition, we stated in our manuscript that CFS showed significant criterion-related validity with well-established but complicated and time-consuming cognitive impairment scales such as MMSE or ABC-DS, including in the Conclusions section. We did not comment that CFS should be used as a dementia scale instead of these scales because of its significant correlations, in the Discussion or in the Conclusions. We appreciate Reviewer 2’s understanding.

---

## [Editor Report · Decision Letter 1]

17 Oct 2022

PONE-D-22-10733R1

Criterion-related validity of Bedriddenness Rank with other established objective scales of ADLs, and Cognitive Function Score with those of cognitive impairment, both are easy-to-use official Japanese scales: a prospective observational study

PLOS ONE

Dear Dr. Tago,

Thank you for submitting your manuscript to PLOS ONE. After careful consideration, we feel that it has merit but does not fully meet PLOS ONE’s publication criteria as it currently stands. Therefore, we invite you to submit a revised version of the manuscript that addresses the points raised during the review process.

Although this revision addresses many of the reviewers' comments, there are a few outstanding issues that require further clarification.

Recommended changes: 

1. In the Methods section, consider incorporating the response provided to Reviewer 1 comment #5. This will clarify, for the reader, why the C domain of the ABC-DS, which would be a more direct measure of cognitive function, was not used as a validation measure in this study.

Required changes:

2. In the Results section, provide some context for why the sensitivity analysis was conducted.

3. In addition, and related to point #2 above, an interpretation of the sensitivity analysis results should be provided in the Discussion section. As it stands right now, neither the rationale for the sensitivity analysis nor the interpretation of its results are provided, potentially leaving the reader wondering why the analysis was carried out and what it accomplished.  

4. The authors should more thoroughly address the first issue raised by Reviewer 2, providing not just a breakdown of diseases leading to hospitalization in the sample but also their potential impact, if any, on study findings. 

5. In addition, revise Table S2 to provide a description of each presented ICD-10 code category (e.g., E codes (Endocrine, nutritional and metabolic diseases)).  

This Academic Editor provides the following disclosure “I participated as a reviewer for the initial evaluation of this manuscript.”

We look forward to receiving your revised manuscript.

Kind regards,

Magdalena Ioana Tolea

Guest Editor

PLOS ONE

Journal Requirements:

Please review your reference list to ensure that it is complete and correct. If you have cited papers that have been retracted, please include the rationale for doing so in the manuscript text or remove these references and replace them with relevant current references. Any changes to the reference list should be mentioned in the rebuttal letter that accompanies your revised manuscript. If you need to cite a retracted article, indicate the article’s retracted status in the References list and also include a citation and full reference for the retraction notice.
---

## [Author Response · Author response to Decision Letter 1]

23 Oct 2022

We thank the editor for reviewing our manuscript. We responded to the editor's comments as follows.

1. In the Methods section, consider incorporating the response provided to Reviewer 1 comment #5. This will clarify, for the reader, why the C domain of the ABC-DS, which would be a more direct measure of cognitive function, was not used as a validation measure in this study.

Response:

As we had explained in the previous revision, only the total score of the ABC-DS is currently available, because of imperfect records by nurses. Therefore, we added the explanation to the Methods section at Line 152-153 on Page 10, as described below.

At Line 152-153 on Page 10

Only the data of total score of ABC-DS was available.

2. In the Results section, provide some context for why the sensitivity analysis was conducted.

Response:

We explained why we conducted the sensitivity analysis in the Results section at Line 215-220 on Page 16, as described below.

At Line 215-220 on Page 16

Additionally, we conducted a sensitivity analysis focusing on the group of patients, for whom MMSE was evaluated because their CFS was abnormal, to improve comparability of the validation analyses between CFS and MMSE, and CFS and ABC-DS. As a result, Spearman’s rank correlation coefficient of CFS with those patients was −0.698 (p < 0.001).

3. In addition, and related to point #2 above, an interpretation of the sensitivity analysis results should be provided in the Discussion section. As it stands right now, neither the rationale for the sensitivity analysis nor the interpretation of its results are provided, potentially leaving the reader wondering why the analysis was carried out and what it accomplished. 

Response:

We provided the interpretation of the sensitivity analysis results in the Discussion section at Line 274-277 on Page 19, as described below.

At Line 274-277 on Page 19 

Furthermore, Spearman’s rank correlation coefficients of CFS with ABC-DS in the group of patients for whom MMSE was evaluated because their CFS was abnormal, revealed a strong negative correlation, with a correlation coefficient of r = −0.698 (p < 0.001).

4. The authors should more thoroughly address the first issue raised by Reviewer 2, providing not just a breakdown of diseases leading to hospitalization in the sample but also their potential impact, if any, on study findings. 

Response:

We conducted a sensitivity analysis by ICD-10 category, as the editor indicated. For categories with sufficient sample size (BR: C, I, J, and K, CFS: I and J), the correlation coefficients between BR or CFS and other scales were similar to those of the overall cohort in this study. On the other hand, there regrettably were some categories (BR: F and H, CFS: B, C, D, R, and S) with insufficient sample sizes resulting in low correlation coefficients, which we considered unreliable. Furthermore, because the validation of correlations between BR or CFS and other scales for each disease category is quite different from the main objective of this study, and the analysis results were unreliable, we decided not to add those results in the main text. In addition, we had already described this point in the Limitation section at Line 305-308 on Page 21, “Thus, the current study may have been affected by population bias caused by the limited number of hospitals, clinical departments, kinds of diseases, and unique characteristics of the region to which the hospital belonged.” We hope the editor understands our intention.

5. In addition, revise Table S2 to provide a description of each presented ICD-10 code category (e.g., E codes (Endocrine, nutritional and metabolic diseases)). 

Response:

We added the specific categories of ICD-10 to S2, Table, following the editor's suggestion.

---

## [Editor Report · Decision Letter 2]

31 Oct 2022

Criterion-related validity of Bedriddenness Rank with other established objective scales of ADLs, and Cognitive Function Score with those of cognitive impairment, both are easy-to-use official Japanese scales: a prospective observational study

PONE-D-22-10733R2

Dear Dr. Tago,

We’re pleased to inform you that your manuscript has been judged scientifically suitable for publication and will be formally accepted for publication once it meets all outstanding technical requirements.

Kind regards,

Magdalena Ioana Tolea

Guest Editor

PLOS ONE
---

## [Editor Report · Acceptance letter]

2 Nov 2022

PONE-D-22-10733R2 

Criterion-related validity of Bedriddenness Rank with other established objective scales of ADLs, and Cognitive Function Score with those of cognitive impairment, both are easy-to-use official Japanese scales: a prospective observational study 

Dear Dr. Tago:

I'm pleased to inform you that your manuscript has been deemed suitable for publication in PLOS ONE. Congratulations! Your manuscript is now with our production department. 

Kind regards, 

on behalf of

Dr. Magdalena Ioana Tolea 

Guest Editor

PLOS ONE